# Sociodemographic, Anthropometric, Body Composition, Nutritional, and Biochemical Factors Influenced by Age in a Postmenopausal Population: A Cross-Sectional Study

**DOI:** 10.3390/metabo13010078

**Published:** 2023-01-03

**Authors:** Héctor Vázquez-Lorente, Lourdes Herrera-Quintana, Jorge Molina-López, Beatriz López-González, Elena Planells

**Affiliations:** 1Department of Physiology, Faculty of Pharmacy, Institute of Nutrition and Food Technology “José Mataix”, University of Granada, 18071 Granada, Spain; 2Faculty of Education, Psychology and Sports Sciences, University of Huelva, 21007 Huelva, Spain

**Keywords:** menopause, aging, nutritional assessment, dietary intake, anthropometry, biochemical parameters

## Abstract

Postmenopausal aging has become relevant for understanding health during the transition life stages—the aging process being involved in several disturbances of the human condition. The present study aimed to investigate the relationship between postmenopausal aging and sociodemographic, anthropometric, body composition, nutritional, and biochemical (i.e., protein and lipid profiles, phosphorous-calcium metabolism, and antioxidant status) factors in postmenopausal women. This cross-sectional study enrolled 78 healthy postmenopausal women (44–76 years). The anthropometrical data showed no differences by age. Biochemical parameters, especially those involved in the protein and phosphorous-calcium metabolism, were influenced by age in our cohort of postmenopausal women. In contrast, no associations were found when considering lipid and antioxidant parameters. Height, fiber intake, blood glucose, protein profile and phosphorous-calcium metabolism markers seem to be the most affected nutritional-related factors by age in our cohort of healthy postmenopausal women. Primary prevention strategies focused on parameters at risk of disruption with postmenopausal aging are necessary to ensure the quality of life in older ages.

## 1. Introduction

Aging has become one of the major issues facing public health, as it promotes multiple diseases and socioeconomic problems [1]. Life expectancy is >80 years for women in developed countries. A better comprehension of effective strategies for ameliorating the aging process will be needed because a third of women’s lives will be spent in the menopausal period [2]. Menopause is considered as the cessation of menstruation permanently, with amenorrhea occurring during 12 successive months before natural menopause is detected, resulting from the loss of ovarian follicular activity directly related to several pathophysiologic conditions [3]. Thus, studies conducted on postmenopausal women have become important to understanding health during the transition stages of life, especially in older women [4].

As menopause advances, women may experience a variety of predictable disorders related to changes in sex hormone levels and aging [5]. In this regard, the decrease in estrogen production often leads to several menopausal symptoms and altered anthropometric and biochemical parameters, particularly those noted at the postmenopausal stage [6]. Similarly, the menopausal transition impacts a wide range of physical and sociodemographic conditions as a result of aging [7].

During post-menopause, women may experience serum lipid changes [8] via the redistribution of the fat mass to the abdominal compartment, reducing activity, energy expenditure, and fat oxidation, thus modifying their body composition via the augmentation of fat mass and diminishing muscle mass [9]. These disturbances in lipid metabolism and anthropometric condition along with an imbalance in antioxidant status increase the risk of impaired cardiovascular and oxidative stress-related parameters as menopause advances [10]. Moreover, bone diseases have been proposed as one of the main factors in postmenopausal stages and are directly related to the aging process. In this regard, age has been inversely correlated with serum calcium (Ca) levels in postmenopausal women, the latter of which are necessary to identify disturbances in phosphorous-calcium (P-Ca) metabolism factors involved in bone disorders, with the purpose of helping to minimize the morbidity/mortality rate and financial burden [11,12]. Physical activity and dietary restrictions have been proposed for menopausal women to help them to face the body composition-related problems mentioned above, these patterns being sometimes modified with aging [13]. As a result of all the previous-mentioned possible changes, postmenopausal women may be at risk of developing several diseases, such as osteopenia and osteoporosis, sexual dysfunction, cardiovascular diseases, cancer, and obesity, among others [14].

As the aging process could affect several factors involved in the modulation of body composition and nutritional factors, the present study aimed to investigate the relationship between postmenopausal aging and sociodemographic, anthropometric, body composition, dietary, and biochemical factors.

## 2. Materials and Methods

### 2.1. Participants and Study Design

The present cross-sectional study was conducted in 78 healthy postmenopausal women aged between 44 and 76 years from the province of Granada (Spain). Women were divided by terciles of age (i.e., tercile 1 = <54 years, tercile 2 = 54–62 years, and tercile 3 = 62 years). All participants signed a written informed consent form. The study was approved by the Ethics Committee of the University of Granada (149/CEIH/2016) and conducted according to the principles of the Declaration of Helsinki [15], in accordance with the International Conference on Harmonization/Good Clinical Practice Standards. Inclusion criteria were (i) to accept to participate in the study after being informed about it, (ii) to present natural postmenopausal status (with at least 12 months of amenorrhea), and (iii) to have healthy condition based on a previous routine hospital laboratory analysis (i.e., normal routinary biochemical parameters, namely glucose, protein, and lipid profiles). Exclusion criteria were (i) to refuse to participate in the study, (ii) to take mineral and vitamin supplements, (iii) to present any pathological condition that could affect their nutritional status (i.e., celiac disease, the main components of metabolic syndrome, bulimia or anorexia), (iv) to be treated with hormone replacement therapy, (v) to have systemic inflammatory status (including C-reactive protein as a reference biomarker to assess the inflammatory status of the patients), and (vi) to present menopause due to non-natural reasons (e.g., surgery or cancer).

### 2.2. Sociodemographic Data Collection

Data regarding physical activity was set as sedentary/non-sedentary, classifying as non-sedentary any participant who performed less than 30 min a day and less than 3 days/week of regular exercise. Regarding smoking habits (non-smoker/smoker), a smoker was a participant who smoked more than one cigarette a day. For educational level (basic educational level/secondary or high educational level), those who only finished primary studies were considered to have a basic educational level. All previous-mentioned data were retrieved using manual questionnaires administered by the researcher. Blood pressure classified as (normal/high pressure) was measured with an electronic sphygmomanometer (HBP-9020, OMRON Co. Ltd., Kyoto, Japan) three times at 30 s intervals in seated participants after resting for 10 min—the mean value being used for further analysis. Values above 80/120 mmHg, for diastolic and systolic pressures, respectively, were considered high pressure values.

### 2.3. Nutrient Intake

The assessment of dietary nutrient intake was performed by a qualified dietitian using a manual 72 h-recall, including one weekend day and two non-holidays. A set of photographs of prepared foods and dishes that are usually consumed in Granada (Spain) were employed for recall accuracy. Dietowin software (7.1. version, Barcelona, Spain) was used to quantitatively convert food intake to energy, carbohydrates, fats, proteins, and fiber content. Adequacy according to the Recommended Dietary Allowance (RDA) for menopausal women from Spain was also determined [16].

### 2.4. Anthropometric and Body Composition Analysis

The participants’ body weight (Kg), and fat free mass (%), were assessed by bioelectrical impedance (Tanita MC-980 Body Composition Analyzer MA Multifrequency Segmental, Barcelona, Spain). The device meets the applicable European standards (93/42EEC, 90/384EEC) for use in the medical research. Height was measured with a stadiometer (Seca, model 213, range 85 to 200 cm; precision: 1 mm; Hamburg, Germany). BMI was calculated as weight (kg)/height (m^2^). The required conditions prior to the measurement were as follows: (i) no alcohol consumption the previous 24 h, (ii) no food or drink intake less than 3 h before, (iii) no vigorous exercise the previous 12 h, and (iv) no urination immediately before the analysis. Waist perimeter was calculated at the midpoint between the top of the iliac crest and the lower margin of the least palpable rib [17]. Hip perimeter was measured at the widest portion of the buttocks, with the tape parallel to the floor [18]. The calculation of waist/hip ratio was made by dividing waist perimeter (cm) by hip perimeter (cm).

### 2.5. Sample Processing

Blood samples were drawn in the morning in fasting conditions and centrifugated at 4 °C for 15 min at 3000 rpm, thus obtaining plasma samples, which were stored at −80 °C. The measurements were performed in the same assay batch with blinded quality control samples being included in the assay batches, in one run.

#### Measurement of Biochemical Parameters

All biochemical factors were measured in plasma. Glucose, creatinine, urea, uric acid, total bilirubin, total proteins, albumin, prealbumin, transferrin, triglycerides, C-reactive protein (CRP), triglycerides, high density lipoprotein (HDL), low density lipoprotein (LDL), total cholesterol, osteocalcin, parathyroid hormone (PTH), and leptin levels were analyzed at the Virgen de las Nieves Hospital (Analysis Unit), Granada (ECLIA, Elecsys 2010 and Modular Analytics E170, Roche Diagnostics, Mannheim, Germany). The reference values were obtained from the Analysis Unit. Flame atomic absorption spectrophotometry (FAAS, Perkin Elmer^®^ Analyst 300 model, Berlin, Germany) was used to analyze Ca, iron (Fe) and copper (Cu), whereas the Fiske-Subbarow colorimetric method (Thermo Scientific, Rockford, IL, United States) was employed to determine phosphorous (P). Vitamin D was analyzed by liquid chromatography–tandem mass spectrometry (LC-MS/MS) using the Waters Acquity UHPLC I-Class System chromatograph (Waters, London, UK). Total antioxidant capacity (TAC) was measured via the reduction power of Cu^2+^ from the action of antioxidants present in plasma samples (TAC kit, Jaica, Shizuoka, Japan). An enzymatic immunological method (Bioxytech GPx-340™ kit, OxisResearch™, Portland, Oregon, USA) was used to determine glutathione peroxidase (GPx) activity. Superoxide dismutase (SOD) activity was analyzed by the colorimetric method based on cytochrome c reduction using the Ransod kit (RANDOX Laboratories Ltd., Crumlin, UK). The rest of the previous-mentioned parameters and their reference values were provided by the Scientific Instrumentation Center (SIC) from the University of Granada.

### 2.6. Statistical Analysis

As a previous step to the execution of a parametric model or not, a Kolmogorov-Smirnov test was used to accept the hypothesis of normal distribution. Frequency analysis for categorical variables was shown as frequencies (*n*) and %. Descriptive analysis for continuous variables was shown as mean and standard deviation (X ± SD). For the comparative analysis based on inter-three groups, one-way ANOVA adjusted by Bonferroni post hoc analysis was employed. Simple linear regression models were used to evaluate the association of age with the significant biochemical parameters of the study. Multiple linear regression models were conducted to test these associations after adjusting for the significant sociodemographic, anthropometric, body composition, and dietary parameters (i.e., smoking habits, educational level, height, and fiber intake). A *p* value < 0.05 was considered significant. All analyses were carried out using the SPSS 22.0 software for Mac (SPSS Inc. Chicago, IL, USA).

## 3. Results

Table 1 shows the anthropometric, body composition, dietary, and sociodemographic variables of the study by age. Regarding anthropometric and body composition variables, only height decreased in those women above 62 years of age compared to postmenopausal women below 54 years of age (*p* < 0.05). In contrast, the rest of the parameters showed no mean differences by age (all at *p* > 0.05). Regarding quantitative dietary intakes, fiber showed the highest values in women older than 62 years of age compared to those younger than 54 years of age (*p* < 0.001), whereas no mean differences by age were reflected for the rest of the studied intakes. In the case of sociodemographic characteristics, smoking habits decreased, and educational level was lower with higher ages (*p* < 0.05 and *p* < 0.01, respectively).

Table 2 represents the biochemical parameters of the study by age. Regarding glucose and protein metabolism parameters, glucose (*p* < 0.001), urea (*p* < 0.05), uric acid (*p* < 0.01) and total bilirubin (*p* < 0.05) increased significantly in women above 62 years of age compared to those below 54 years of age. Moreover, the lipid parameters of our study showed no mean differences by age (all *p* > 0.05). Regarding the parameters involved in P-Ca metabolism, P (*p* < 0.01), vitamin D (*p* < 0.01) and vitamin D_3_ metabolite (*p* < 0.05) levels were significantly higher in women aged 54–62 years of age than in those aged < 54 years of age, whereas Cu levels was significantly lower in women above 62 years of age than in those < 54 years of age (*p* < 0.001). The rest of the parameters showed no mean differences by age group (*p* > 0.05).

Table 3 shows the significant associations between age and the biochemical parameters of the study (Model 0), adjusted for the sociodemographic, anthropometric, body composition and dietary parameters showing differences by age in Table 1. Age showed a direct correlation with glucose, uric acid, and osteocalcin (all *p* ≤ 0.032; Model 0), and a negative relationship with albumin and Cu (all *p* ≤ 0.038; Model 0), which persisted after including smoking habits, educational level, height, and fiber intake as covariates (all *p* ≤ 0.042; Model 1). Age additionally showed a direct relationship with creatinine, urea, and total bilirubin (All *p* ≤ 0.011; Model 0), which lost their significance after adjusting by the covariates mentioned above (All *p* ≥ 0.072; Model 1).

## 4. Discussion

This study tried to elucidate the influence of aging upon several sociodemographic, anthropometric, body composition, dietary, and biochemical factors in a cohort of postmenopausal women. Both educational level and smoking habits tended to be lower with aging, and contrary to our primary hypothesis, the analyzed anthropometric and body composition data showed no differences by age. In contrast, biochemical parameters, especially those involved in the protein and P-Ca metabolism, were influenced by age, whereas such associations were not found when including lipid and antioxidant parameters.

Regarding anthropometric and body composition parameters, no differences by age were found except for height, which was significantly lower in the group with greater age compared to the youngest group, with BMI showing no intergroup differences, which was considerably high (around 27 kg/m^2^) in the three groups. Some authors have reported that height decreases with age, whereas BMI and weight significantly increase at the same time [19]. This could be due to the loss in both lean mass and height with aging [20], and to the lower physical condition at older ages [21]. Aging is independently correlated with increasing BMI [22], and significantly higher waist circumference values were found with higher ages [23]. Postmenopausal women tend to present impaired body composition, energy expenditure, or insulin sensitivity compared to similarly overweight premenopausal women [24].

In the present study, we observed a lower educational level with advancing age. Educational level is considered one of the best socioeconomic indicators, and its relation to menopause has been considered by some authors [25]. Older people, having less access to education than their younger counterparts, present lower educational attainment, which could affect their health [26]. In addition, smoking habits decreased significantly, and physical activity showed no differences with aging in our population of postmenopausal women. It has been observed that low physical activity and smoking habits are less prevalent in postmenopausal women than in premenopausal and climacteric women [27]. Concerning blood pressure, no statistically significant differences by age were found. Counterintuitively, menopause is considered to be the major determinant of blood pressure increase in women [28].

The biochemical variables analyzed in the present study showed differences by age in several parameters. Plasma glucose levels were higher at greater ages, but lipid parameters showed no significant differences. Aging has been proposed to be a risk factor in developing insulin resistance, thus increasing circulating glucose levels [29]. Among women, menopause is associated with a more atherogenic lipid profile [30]. Total cholesterol, triglycerides, and the total cholesterol/HDL ratio increase from premenopausal to postmenopausal status [31]. However, in middle-aged women, age does not seem to modulate such changes in most of the studied metabolic health indicators during the menopausal transition (i.e., glucose, triglycerides, total cholesterol, HDL, and LDL) [32]. With aging, we found a significant increase in protein metabolism (i.e., urea, uric acid, and bilirubin). The influence of age on urea levels has been previously addressed in a population of 1000 middle-aged (40–60 years) women (50% postmenopausal), circulating urea levels being higher in postmenopausal women than in women in the menopausal transition and premenopausal stages [33]. Other authors have reported decreases in circulating urea levels, whereas circulating total proteins and albumin are higher in postmenopausal women at 60 years of age than in women at premenopausal stages at 42 years of age [34]. Age is one of the main contributors to increased circulating uric acid levels [35]. Blood urea nitrogen, creatinine, and uric acid are circulating markers of renal function whose values are elevated when renal function is reduced, especially in aging [36]. In this regard, we have observed a significant association between age and urea, uric acid, and creatinine, with uric acid maintaining its significance after the adjustment of the above-mentioned covariates, which could indicate a deterioration in the renal function at greater ages.

The homeostasis of Ca and P is maintained by a concerted interplay of absorption and resorption and by storage and mobilization from the bone, regulated mainly by PTH, vitamin D, and calcitonin [37]. In this regard, we found higher mean vitamin D levels in women between 54–62 years than in those below 54 years, observing the same trend for P levels. However, no statistically significant mean differences were found for osteocalcin, PTH, and Ca, which is in contrast with the decreased serum Ca levels with increasing age in 252 Nepalese postmenopausal women reported by Pardhe et al. [11]. Age appears to be an important independent predictor of serum Vitamin D in this population [38]. A study conducted on ninety-nine women aged ≥50 years showed higher vitamin D levels at higher ages in postmenopausal women [39]. Other authors have found low vitamin D levels in women whose age was below 48 and over 60 [40]. It must be noted that the low levels of vitamin D that we obtained are mainly due to low levels of the 25-OH-D_3_ metabolite. The supplementation with Ca and vitamin D_3_ may normalize 25-OH-D_3_ levels in postmenopausal women with inadequate circulating levels of 25-OH-D_3_ and could reduce circulating PTH levels [41]. Furthermore, we observed a significant direct association of age with osteocalcin, which persisted after adjusting for smoking habits, educational level, height, and fiber intake. Osteocalcin is a small bone-specific non-collagen protein produced by osteoblasts, being a sensitive marker of bone formation [42]. Osteocalcin is also involved in energy metabolism and preventing age-related muscle loss [43]. Age and years since menopause have been reported to be significantly associated with serum osteocalcin levels and bone mineral density [44], age being one of the most important predictors of osteocalcin status in postmenopausal women [45]. Interestingly, higher plasma osteocalcin levels have been found in postmenopausal compared to perimenopausal women [46].

Healthy aging is particularly important in women, as their lifespan is generally longer than that of men, leaving women at higher risk for age-related diseases [47]. Our findings may contribute to the understanding of processes of womens’ healthy aging and the development of interventions targeting these indicators, which could have significant relevance for public health.

The present study suffers from some limitations, including that (i) we had no information about time living with menopause, which could have helped to enrich the way to interpret age-related results, (ii) it has a cross-sectional design, which means that no causal relationships can be established, (iii) we recruited fewer participants than desired and therefore the sample size of the study may not be big enough to obtain more statistically significant results, (iv) the study population was limited to postmenopausal women aged between (44–76 years old) from a specific area of southern Spain, and hence these results may not be generalizable to postmenopausal women of different regions or with ages not included in our range of age, and (v) it would have been interesting to include women in the premenopausal period to compare the effect of aging upon the studied parameters in women of different menopausal stages. Moreover, further studies including larger sample sizes and more parameters should additionally be addressed.

## 5. Conclusions

In conclusion, height, fiber intake, blood glucose, protein and P-Ca metabolism parameters were the main variables that seem to be altered with the aging process. In contrast, anthropometric, lipid, and antioxidant parameter values showed no changes with aging. Further studies are needed to confirm our results in other postmenopausal populations to establish primary prevention protocols to maintain the variables that tend to be altered with aging in postmenopausal populations.

## Figures and Tables

**Table 1 metabolites-13-00078-t001:** Anthropometric, body composition, dietary, and sociodemographic variables by age.

Characteristics	Total Population (*n* = 78)	<54 Years (*n* = 26)	54–62 Years (*n* = 26)	>62 Years (*n* = 26)	*p* Value	Reference Values
	Mean ± SD	Mean ± SD	Mean ± SD	Mean ± SD		
**Anthropometric and body composition parameters**						
Weight (Kg)	68.7 ± 13.2	71.1 ± 15.1	68.5 ± 12.3	66.2 ± 11.8	0.414	
Height (m)	159.3 ± 6.23	162.1 ± 6.35 ^a^	158.9 ± 5.68	156.5 ± 5.43 ^a^	0.004	
BMI (Kg/m^2^)	27.0 ± 4.60	26.9 ± 5.07	27.1 ± 4.52	27.0 ± 4.31	0.996	22.0–27.0
Waist perimeter (cm)	89.0 ± 12.6	87.4 ± 16.6	88.9 ± 12.4	91.0 ± 13.3	0.592	<90.0
Hip perimeter (cm)	105.8 ± 10.5	105.9 ± 9.61	106.5 ± 10.7	104.9 ± 11.5	0.873	<110.0
Waist/hip ratio	0.83 ± 0.08	0.81 ± 0.08	0.83 ± 0.07	0.85 ± 0.08	0.202	<0.80
Fat mass (%)	37.6 ± 5.92	36.9 ± 5.77	38.6 ± 5.66	37.2 ± 6.46	0.556	23.0–31.0
Fat free mass (%)	62.4 ± 5.92	63.1 ± 5.77	61.4 ± 5.66	62.7 ± 6.46	0.556	>69.0
**Dietary intake**						
Energy (kcal/day)	1378 ± 337	1361.8 ± 394.4	1326.4 ± 304.7	1457.8 ± 294.6	0.381	2000.0
Carbohydrates (g/day)	149.7 ± 42.5	144.9 ± 49.2	142.9 ± 40.1	162.7 ± 34.7	0.200	275.0
Fats (g/day)	59.1 ± 20.6	59.9 ± 25.4	56.1 ± 17.1	61.2 ± 18.1	0.658	70.0
Proteins (g/day)	61.6 ± 15.4	59.2 ± 14.3	60.3 ± 15.4	65.7 ± 16.3	0.281	50.0
Fiber (g/day)	15.9 ± 8.11	12.9 ± 5.23 ^a^	14.9 ± 5.83	20.7 ± 10.7 ^a^	0.001	>25
**Characteristics**	***n* (%)**	***n* (%)**	***n* (%)**	***n* (%)**	***p* Value**	**Reference Values**
**Sociodemographic**						
Blood pressure	-	-	-	-	-	-
Normal blood pressure	43 (55)	18 (69)	14 (54)	11 (42)	0.165	-
High blood pressure	35 (45)	8 (31)	12 (46)	15 (58)		-
**Physical exercise**	-	-	-	-	-	-
Sedentary	20 (26)	9 (35)	5 (19)	6 (23)	0.310	-
Non-sedentary	58 (74)	17 (65)	21 (81)	20 (77)		-
**Smoking habit**	-	-	-	-	-	-
Non-smoker	62 (80)	17 (65)	23 (89)	22 (85)	0.045	-
Smoker	16 (20)	9 (35)	3 (11)	4 (15)		-
**Educational level**	-	-	-	-	-	-
Basic educational level	29 (37)	7 (27)	6 (23)	16 (62)	0.008	-
Secondary or high educational level	49 (63)	19 (73)	20 (77)	10 (38)		-

*n* = 78. Categorical variables are expressed as frequencies (*n*) and percentages (%). Continuous variables are indicated as mean ± standard deviation (SD). To compare inter-tercile groups’ categorical variables, ^a^ Chi-square test was used. To compare inter-tercile groups’ continuous variables, one-way ANOVA with adjustment by Bonferroni post hoc test analysis was used: ^a^ = < 54 years vs. >62 years. Significance was set at a *p* value < 0.05. Abbreviations: BMI = Body Mass Index.

**Table 2 metabolites-13-00078-t002:** Biochemical variables by age.

Characteristics	Total Population (*n* = 78)	<54 Years (*n* = 26)	54–62 Years (*n* = 26)	>62 Years (*n* = 26)	*p* Value	Reference Values
	Mean ± SD	Mean ± SD	Mean ± SD	Mean ± SD		
Glucose (mg/dL)	92.2 ± 15.9	85.0 ± 12.9 ^b^	91.6 ± 9.31	100.9 ± 20.2 ^b^	0.001	70.0–110.0
Creatinine (mg/dL)	0.69 ± 0.13	0.66 ± 0.10	0.68 ± 0.07	0.74 ± 0.18	0.085	0.50–0.90
Urea (mg/dL)	34.5 ± 9.08	32.9 ± 0.09	32.4 ± 6.97 ^c^	38.7 ± 10.5 ^c^	0.024	10.0–50.0
Uric acid (mg/dL)	4.40 ± 1.07	4.10 ± 0.90 ^b^	4.20 ± 0.96 ^c^	4.97 ± 1.16 ^b,c^	0.005	2.40–5.70
Total bilirubin (mg/dL)	0.47 ± 0.14	0.42 ± 0.12 ^b^	0.48 ± 0.10	0.53 ± 0.16 ^b^	0.023	0.10–1.20
Total proteins (g/dL)	7.08 ± 0.52	7.12 ± 0.60	7.07 ± 0.45	7.05 ± 0.53	0.883	6.60–8.70
Albumin (mg/dL)	4.44 ± 0.21	4.49 ± 0.22	4.48 ± 0.22	4.37 ± 0.18	0.120	3.50–5.20
Prealbumin (mg/dL)	25.2 ± 5.07	24.9 ± 4.10	25.8 ± 5.53	24.8 ± 5.87	0.805	20.0–40.0
Transferrin (mg/dL)	280.2 ± 45.9	287.5 ± 42.5	269.1 ± 45.9	283.4 ± 49.9	0.385	200.0–360.0
CRP (mg/L)	1.04 ± 6.95	2.35 ± 11.3	0.32 ± 0.43	0.22 ± 0.13	0.447	0.02–5.00
Triglycerides (mg/dL)	108.2 ± 67.9	102.5 ± 84.2	109.6 ± 62.8	113.4 ± 52.3	0.845	50.0–200.0
HDL (mg/dL)	66.6 ± 15.6	67.5 ± 13.1	65.2 ± 10.1	67.3 ± 22.3	0.845	40.0–60.0
LDL (mg/dL)	128.0 ± 31.3	122.2 ± 29.0	137.8 ± 31.2	124.4 ± 32.9	0.156	70.0–190.0
Total cholesterol (mg/dL)	220.5 ± 34.4	215.4 ± 31.0	227.2 ± 36.7	219.3 ± 35.8	0.453	110.0–200.0
Osteocalcin (ng/mL)	15.3 ± 9.82	13.1 ± 8.96	15.3 ± 8.45	18.0 ± 11.7	0.209	15.0–46.0
PTH (pg/mL)	56.2 ± 23.8	57.6 ± 31.0	51.7 ± 15.6	59.3 ± 20.7	0.516	20.0–70.0
Leptin (ng/mL)	13.9 ± 4.83	13.4 ± 5.02	14.9 ± 5.28	13.4 ± 4.07	0.400	3.60–11.1
25–OH–D (ng/mL)	23.5 ± 7.40	20.7 ± 5.49 ^a^	27.1 ± 8.30 ^a^	22.8 ± 6.98	0.006	30.0–100.0
25–OH–D_3_ (ng/mL)	17.7 ± 7.06	15.1 ± 5.39 ^a^	20.9 ± 7.90 ^a^	17.5 ± 6.74	0.012	>20
25–OH–D_2_ (ng/mL)	5.74 ± 3.11	5.62 ± 2.32	6.25 ± 4.27	5.31 ± 2.41	0.585	>10
Ca (mg/dL)	9.21 ± 0.44	9.13 ± 0.32	9.26 ± 0.42	9.24 ± 0.58	0.531	8.60–10.2
P (mg/dL)	3.49 ± 0.50	3.25 ± 0.57 ^a^	3.70 ± 0.36 ^a^	3.54 ± 0.42	0.003	2.70–4.50
Fe (µg/dL)	92.6 ± 30.7	83.1 ± 31.8	100.2 ± 29.0	95.7 ± 29.5	0.109	60.0–170.0
Cu (µg/dL)	101.4 ± 23.0	111.8 ± 24.3 ^b^	101.4 ± 20.6	85.3 ± 15.4 ^b^	0.001	85.0–180.0
TAC (µmol/L)	1539.3 ± 483.1	1585.0 ± 658.3	1405.5 ± 388.1	1634.9 ± 264.1	0.209	1500.0
GPX (mU/mL)	118.2 ± 47.7	117.4 ± 43.9	116.9 ± 37.6	120.7 ± 62.1	0.957	120.0
SOD (U/mL)	184.4 ± 34.2	181.6 ± 33.1	184.3 ± 35.2	187.8 ± 35.5	0.813	164.0–240.0

*n* = 78. All variables are expressed as mean ± standard deviation (SD). To compare inter-tercile groups’ continuous variables, one-way ANOVA with adjustment by Bonferroni post hoc test analysis was used: ^a^ = < 54 years vs. 54–62 years, ^b^ = < 54 years vs. > 62 years, and ^c^ = tercile 54–62 years vs. > 62 years. Significance was set at a *p* value < 0.05. Abbreviations: Ca = Calcium; Cu = Copper; CRP = C-Reactive Protein; Fe = Iron; HDL = High Density Lipoprotein; LDL = Low Density Lipoprotein; P = Phosphorous; PTH = Parathyroid Hormone; TAC = Total Antioxidant Capacity; GPx = Glutathione Peroxidase; SOD = Superoxide Dismutase.

**Table 3 metabolites-13-00078-t003:** Significant associations between age and the biochemical parameters of the study.

Characteristics	Model 0			Model 1		
	ß	R^2^	*p* Value	ß	R^2^	*p* Value
Glucose (mg/dL)	0.425	0.181	0.001	0.377	0.237	0.007
Creatinine (mg/dL)	0.298	0.089	0.009	0.273	0.100	0.072
Urea (mg/dL)	0.307	0.094	0.007	0.169	0.133	0.252
Uric acid (mg/dL)	0.404	0.163	0.001	0.391	0.167	0.008
Total bilirubin (mg/dL)	0.292	0.085	0.011	0.162	0.186	0.260
Albumin (mg/dL)	−0.243	0.059	0.038	−0.398	0.121	0.011
Cu (µg/dL)	−0.379	0.144	0.002	−0.347	0.160	0.042
Osteocalcin (ng/mL)	0.247	0.061	0.032	0.436	0.130	0.004

ß = standardized regression coefficient. R^2^ and *p* are from simple and multiple regression analyses between age and the significant biochemical parameters: Model 0 = simple regression analysis; Model 1 = Multiple regression analysis adjusted by significant sociodemographic, anthropometric, body composition and dietary parameters which showed differences by age (i.e., smoking habits, educational level, height, and fiber intake). Significance was set at *p* value < 0.05. Abbreviations: Cu = Copper.

## Data Availability

All datasets generated for this study are included in the article.

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
