# Peer review of "Sociodemographic, Anthropometric, Body Composition, Nutritional, and Biochemical Factors Influenced by Age in a Postmenopausal Population: A Cross-Sectional Study"

_metabolites, 2023, doi:10.3390/metabo13010078_

Round 1
Reviewer 1 Report
1. The reviewer couldn't find any novelty of the manuscript. Most of the data were related to aging, not menopause. The author described what happens during menopause in intro, so, to study that question, the population should include the women in menopausal transition, not completely done in menopause.
2. The number of women recruited in the study was small. As this is a cross-sectional study, more subject numbers are required or the author might want to use different statistical methods to overcome the limitation.
Author Response
Comments from the Editors and Reviewers:
First of all, we are very grateful to the reviewer for the suggestions made, which, we believe, have contributed to the remarkable improvement of our manuscript. The following is a list of each and every one of the responses to the proposed comments.
Reviewer #1:
Point 1:
The reviewer couldn't find any novelty of the manuscript. Most of the data were related to aging, not menopause. The author described what happens during menopause in intro, so, to study that question, the population should include the women in menopausal transition, not completely done in menopause.
Response 1:
Dear reviewer, the authors focused the study to assess the effect that the aging process could have on various sociodemographic, anthropometric, body composition, nutritional and biochemical parameters in women who are already in postmenopausal period. When designing the study, we included only postmenopausal women to avoid other confounding factors that may be present in premenopausal women, such as better hormone status, among others. Based on this, and the appropriate suggestions from the reviewer, we have modified several parts of the entire manuscript to make it clear that we are assessing the effect of age on women at a particular stage of their lives such as the postmenopausal period with the intention of valuating primary prevention protocols to avoid the studied parameters alterations in concrete moments of women’s’ life. In addition, we have added in limitations the possibility of including a comparison with women in the premenopausal process in future studies. We appreciate the reviewer’s comment.
Point 2:
The number of women recruited in the study was small. As this is a cross-sectional study, more subject numbers are required, or the author might want to use different statistical methods to overcome the limitation.
Response 2:
Dear reviewer, we do not currently have the necessary funding to increase the sample size in our study. On the other hand, based on our number of participants, we applied the normality tests on this basis corresponding to parametric tests. However, we have clearly defined that the small sample size is a clear limitation of the study in the ‘discussion’ section.
Reviewer 2 Report
It is an interesting study regarding the factors influenced by age in postmenopausal women.
The title should be more descriptive of the factors studied, since there was also a sector regarding anthropometric and body composition analysis, unless the authors suggest that all body composition changes are of nutritional origin.
The introduction should describe the health risks that this changes impose.
The conclusion should make a reference on all the factors that seem to show statistical difference due to advanced age, or explain why some of them are not included.
Some minor English language and style would be of use.
Author Response
Comments from the Editors and Reviewers:
First of all, we are very grateful to the reviewer for the suggestions made, which, we believe, have contributed to the remarkable improvement of our manuscript. The following is a list of each and every one of the responses to the proposed comments.
Reviewer #2:
It is an interesting study regarding the factors influenced by age in postmenopausal women.
Point 1:
The title should be more descriptive of the factors studied, since there was also a sector regarding anthropometric and body composition analysis, unless the authors suggest that all body composition changes are of nutritional origin.
Response 1:
According to reviewer’s appropriate suggestion, we have expanded the tittle of this manuscript to give a better approach of the parameters analyzed in the study.
Point 2:
The introduction should describe the health risks that this changes impose.
Response 2:
Thank you for the suggestion, we have included this information in the ‘introduction’ section.
Point 3:
The conclusion should make a reference on all the factors that seem to show statistical difference due to advanced age, or explain why some of them are not included.
Response 3:
Dear reviewer, as suggested, we have provided more information about all these factors in the conclusion section and abstract.
Point 4:
Some minor English language and style would be of use.
Response 4:
Thank you for the suggestion, we have carefully evaluated once again English language and style with a native speaker translator.
Reviewer 3 Report
Dear authors.
Thank you for writing this article.
Overall, the topic is interesting on the association between aging and different factors in postmenopausal women. However, there are major comments needed to be considered:
- A similar previous study by Pardhe et al. (10.2147/IJWH.S145191) showed that age was significantly correlated with calcium in postmenopausal women. It is better to cite this article in the introduction and clarify the novelties of your work. Please also bring some limitations of your work when comparing this study in the discussion (such as sample size and etc.)
- The sample size of your study is very small to reach a strong conclusion in a cross-sectional study design. That would definitely be the main reason for not observing any significant association.
- The definition of postmenopausal women needed to be more clarified; i.e. only women with primary (natural) menopause were included, or women who had menopause due to other reasons such as surgery, and etc. were also included? If yes, how did you manage that? Secondary reasons for menopause would definitely affect the final analyses, and further adjustments are needed.
- What types of test did you mean by “on a previous routine hospital laboratory analysis”, in the inclusion criteria. The eligibility criteria needs to be more specified.
- Why did you not consider energy intake as covariate for final analyses?
- Why did you not evaluate the age relation with body fat% or LBM as the anthropometric parameters? Baseline measures were also reported in the study!
- More precise limitations, as well as suggestions for future studies must be brought in your study.
Author Response
Comments from the Editors and Reviewers:
First of all, we are very grateful to the reviewer for the suggestions made, which, we believe, have contributed to the remarkable improvement of our manuscript. The following is a list of each and every one of the responses to the proposed comments.
Reviewer #3:
Dear authors.
Thank you for writing this article.
Overall, the topic is interesting on the association between aging and different factors in postmenopausal women. However, there are major comments needed to be considered:
Point 1:
A similar previous study by Pardhe et al. (10.2147/IJWH.S145191) showed that age was significantly correlated with calcium in postmenopausal women. It is better to cite this article in the introduction and clarify the novelties of your work. Please also bring some limitations of your work when comparing this study in the discussion (such as sample size and etc.)
Response 1:
Thank you for the suggestion, we have included this study in the ‘introduction’ and ‘discussion’ section and specify the novelties of this work. Likewise, we have described more limitations in the ‘discussion’ section.
Point 2:
The sample size of your study is very small to reach a strong conclusion in a cross-sectional study design. That would definitely be the main reason for not observing any significant association.
Response 2:
Dear reviewer, we were unable to expand the sample size due to lack of funding. However, we have explained in the limitations section the interesting suggested point that there may be a lack of associations due to a small sample size.
Point 3:
The definition of postmenopausal women needed to be more clarified; i.e. only women with primary (natural) menopause were included, or women who had menopause due to other reasons such as surgery, and etc. were also included? If yes, how did you manage that? Secondary reasons for menopause would definitely affect the final analyses, and further adjustments are needed.
Response 3:
As suggested, we have clarified this information in ‘methods’ section. Thank you for the suggestion.
Point 4:
What types of test did you mean by “on a previous routine hospital laboratory analysis”, in the inclusion criteria. The eligibility criteria needs to be more specified.
Response 4:
Dear reviewer, we have clarified this point in ‘Materials and Methods’ section.
Point 5:
Why did you not consider energy intake as covariate for final analyses?
Response 5:
Dear reviewer, we decided to include in the regression analyses only those variables that presented significant differences based on age after applying the one-way ANOVA test in table 1. Additionally, we made this decision because our regression model might not accept more covariates due to the small sample size.
Point 6:
Why did you not evaluate the age relation with body fat% or LBM as the anthropometric parameters? Baseline measures were also reported in the study!
Response 6:
Dear reviewer, as you suggested, we performed the abovementioned simple linear regression analysis, but regretfully, we did not obtain any statistically significant results. Moreover, as we detected no mean differences of body composition parameters in table 1 by age, we considered not to additionally add the simple linear regression analysis. Thank you for the comment.
Point 7:
More precise limitations, as well as suggestions for future studies must be brought in your study.
Response 7:
According to reviewer’s appropriate suggestion, we have included more precise information in relation to limitations, as well as some suggestions for future studies in ‘discussion’ section.
Round 2
Reviewer 1 Report
The author has significantly improved the manuscript according to reviewer's suggestions.
Reviewer 3 Report
Dear authors,
Thank you for your responses.